# Knowledge building and vocabulary growth: Assessing the impact of seamless Chinese vocabulary learning for international students

**Xiaosheng Zhou**[1]*, **Ying Soon Goh**[2]

1 School of Literature and Media, Wenzhou University of Technology, Wenzhou, China, 2 Akademi Pengajian Bahasa, Universiti Teknologi MARA (UiTM), Campus Terengganu Dungun, Dungun, Terengganu, Malaysia

* seanzhou@whu.edu.cn

## Abstract

This study investigates the effectiveness of Seamless Chinese Vocabulary Learning (SCVL) among international students learning Chinese as a Foreign Language (CFL) to foster vocabulary knowledge building. A new theoretical framework of SCVL was introduced and validated to guide this exploration. The research involved 32 international students enrolled in a Chinese university. Data collection included Chinese Lexical Frequency Profile (LFP) were at HSK Level 4, a SCVL questionnaire, and semi-structured interviews. The findings indicate that SCVL significantly enhances students' HSK Level 4 vocabulary learning and retention across diverse performance groups over time, while also providing a positive and engaging learning experience. SCVL creates an authentic and repetition-enabled learning context, fostering higher levels of interaction and multi-modal, immediate, and learner-friendly scaffolding. Moreover, the study reveals that SCVL motivates students to actively participate in vocabulary acquisition, despite facing certain challenges. By incorporating the SCVL framework, language instructors can enhance their pedagogical practices and promote sustainable language learning outcomes. Future research is recommended to include a broader range of learners with diverse backgrounds and language proficiency levels, as well as to compare seamless learning with traditional learning approaches. Additionally, exploring CFL teachers' perceptions of SCVL would be valuable to further understand its impact on language instruction.

## Introduction

Seamless learning, characterised by its ability to integrate formal and informal learning contexts, represents a transformative approach in education [1,2]. By leveraging mobile and digital technologies, this approach facilitates continuous learning experiences that span across physical and virtual environments, fostering deep engagement and knowledge retention [2,3]. Within this framework, learners are encouraged to integrate real-world practices into their academic development, making seamless learning particularly effective for skill-based disciplines such as language acquisition [2].

**Data availability statement:** The data is stored on openICPSR with the following details: Accession Number: openicpsr-211323 Data Link: https://www.openicpsr.org/openicpsr/workspace?path=openICPSR

**Funding:** The author(s) received no specific funding for this work.

In the domain of second language acquisition, vocabulary development plays a pivotal role. Research highlights that contextualised, meaningful learning activities are essential for fostering vocabulary retention and application [4]. To effectively foster vocabulary acquisition, a crucial prerequisite is the establishment of professional and authoritative vocabulary syllabi that guide the language learning process [2,5]. In this regard, China has made significant strides in supporting international students' Chinese vocabulary acquisition endeavors through the development of the "Syllabus of Chinese Vocabulary and Chinese Characters" in 1991 and the establishment of the "Graded Chinese Syllables, Characters and Words for the Application of Teaching Chinese to the Speakers of Other Languages" as a national standard in 2010.

Furthermore, the Chinese Ministry of Education introduced the Chinese Proficiency Test (HSK or Hanyu Shuiping Kaoshi) as an international standardized test to assess the proficiency of non-native Chinese speakers. The HSK 1.0 Word List was officially launched in 1996, followed by the HSK 2.0 Word List in 2010. Subsequently, in 2021, the HSK 3.0 Word List was updated and partially implemented on July 1, 2021, with the aim of providing comprehensive guidance for future HSK examinations. Table 1 shows the vocabulary size of different new HSK level in HSK 2.0 word list.

During this transitional phase, as recommended by numerous scholars [6,7], the HSK 2.0 Word List continues to serve as the vocabulary syllabus for the six global levels of the Chinese Proficiency Examination. For the purpose of this research, the HSK 2.0 Word List was utilized to focus on specific aspects of vocabulary.

The effort taken by the Chinese government to develop the HSK vocabulary lists reflects the significance of the necessity to master Chinese vocabulary, particularly in the field of teaching and learning CFL. However, learning Chinese as a foreign language presents substantial challenges due to the complexities of its writing system and the extensive lexicon required to communicate [8]. Among the many difficulties faced by CFL learners in China, Chinese writing is particularly challenging, often due to students' limited store of vocabulary [9,10]. Nevertheless, traditional methods of vocabulary teaching often fail to engage learners in ways that connect formal classroom instruction with informal, real-life usage scenarios [2,11,12]. This gap presents an opportunity to apply the principles of seamless learning to bridge the divide between theory and practice in SLA.

In this context, the Seamless Chinese Vocabulary Learning (SCVL) approach addresses this challenge by embedding vocabulary acquisition into authentic, contextualised activities such as essay writing. SCVL leverages the affordances of mobile technologies and collaborative environments to integrate vocabulary practice with real-world applications [12,13]. By doing so, it not only aligns with the principles of seamless learning but also fosters deeper connections between vocabulary knowledge and practical language use.

Consequently, this study placed great emphasis on enhancing the process of learning Chinese vocabulary for international students by utilising the SCVL approach. The aim was to

**Table 1. The vocabulary size of different new HSK level in HSK 2.0 word list.**

| Six Levels of HSK 2.0 Word List | Vocabulary Size |
| --- | --- |
| HSK (Level 6) | Over 5,000 |
| HSK (Level 5) | 2500 |
| HSK (Level 4) | 1200 |
| HSK (Level 3) | 600 |
| HSK (Level 2) | 300 |
| HSK (Level 1) | 150 |

optimise the learning process and aid international students in comprehending and communicating successfully in written Chinese.

However, despite the substantial research investigating the integration of seamless learning approaches and their impact on language skill acquisition, learning strategies, attitudes, and motivation [14–16], there remains a significant gap in qualitative insights into the authentic experiences, emotions, and challenges faced by CFL students during seamless vocabulary learning. For instance, most empirical studies have focused on quantitative measures, such as pretest-posttest designs, to evaluate the effectiveness of mobile applications or learning systems within a seamless learning framework [14,16]. Nonetheless, understanding students' perspectives is essential for identifying both the benefits and limitations of seamless vocabulary learning and for informing instructional strategies [17]. While studies, such as that by [16], have utilized quantitative methods to assess vocabulary acquisition, qualitative insights into learners' experiences are essential for developing a more nuanced and comprehensive understanding of the impact of seamless learning on vocabulary development.

To address this gap, the present study employs the Design-Based Research (DBR) methodology, which aligns closely with the principles of seamless learning [18]. DBR facilitates the iterative refinement of instructional designs, informed by real-world applications, participant feedback, and contextual requirements [19]. This study places particular emphasis on students' learning outcomes within a seamless language learning environment, focusing on authentic and contextualised tasks such as essay writing. While the research does not provide an in-depth exploration of instructional design details, it prioritises an evaluation of students' experiences and measurable learning outcomes [2,13].

As highlighted by [20], DBR is especially effective in examining learners' engagement and knowledge-sharing behaviours in mobile seamless learning contexts. In line with this approach, the present study investigates how authentic writing tasks—specifically, student essay artefacts incorporating HSK Level 4 vocabulary—support vocabulary acquisition and application. By integrating real-world and contextualised tasks, this research underscores the importance of seamless learning in bridging formal and informal educational processes to enhance language proficiency.

The analysis focuses primarily on student essay artefacts, which serve as tangible representations of learners' ability to apply target vocabulary. These artefacts provide valuable insights into the effectiveness of seamless learning environments in fostering both vocabulary acquisition and practical language use.

## Literature review

Seamless learning is an emerging learning notion or approach that refers to a person experiencing continuity of learning across a combination of locations, times, technologies or social settings, and consciously connecting such multifaceted and multi-modal learning efforts to achieve deeper learning [2,3]. Studies have demonstrated the efficacy of seamless learning in promoting student vocabulary learning through photo-taking and sentence making using mobile phones [1–3]. Furthermore, the use of seamless learning in CFL has been found to enhance students' language performance, motivation, interest, and engagement [12,13]. However, previous studies have mainly focused on primary school students and sentence artifact making in Singapore, with limited attention given to self-directed learning due to cognitive load [21,22].

Although seamless learning in L2 has demonstrated promising outcomes in literature, its application in CFL is still in its early stages when compared to other more established technology-enhanced learning models such as web-based platforms and VR/AR-based learning models [23–25]. As a result, the generalisability of findings may be limited due to the

relatively small amount of research conducted thus far [21]. Furthermore, there has been limited research on the adoption of SCVL in authentic educational settings in Chinese mainland [12], underscoring the need for further research to provide evidence of its possible applications in tertiary CFL contexts.

The situated learning theory posits that learning occurs within an authentic context, culture, and activity and that it is widely unintentional [26,27]. Authenticity plays a crucial role in situated learning and successful foreign language learning [28]. Learning in an authentic setting allows students to gain practical experience, formulate and test linguistic hypotheses, and develop language skills in real-life situations [29]. As stated by [30], seamless learning with the support of the situated learning theory allows learners to shift from one learning context to the other through a mere touch on their smart devices. Thus, the target vocabulary words are never presented in isolation. Instead, they are always situated within a context, requiring the learner to utilise contextual clues and their prior knowledge to apply the meanings of the words.

Neo-constructivism is a learning and innovation theory that is suitable for the network age [31]. According to [31], the neo-constructivism learning theory serves as the foundation for personalized learning engagement in the network era. Learners can learn to retrieve and select knowledge under their educators' guidance and learn by utilising 'piece-writing', 'personalised rewriting', and 'creative reconstruction' to gradually build a personalised 'three-level knowledge structure' [31,32]. Therefore, it emphasises that educators should learn to construct situations and learners should learn to search for and choose knowledge meaningfully, undertake efficient communication, and realise innovation and meaning construction [16]. Neo-constructivism learning theory should be employed to support seamless learning, as it facilitates a smooth transition of learning across different mediums [17,21]. For example, when receiving information in the classroom, learners can seamlessly venture to a different medium like a smart device to further explore and expand on the information.

The connectivism theory views learning as a networked process in which knowledge and understanding are acquired through connections and relationships with other people, information sources, and experiences [3]. According to [33], the founder of the theory of connectivism, learning is a process of connecting specialised nodes or information sources. Through the interactive implementation of SCVL using mobile devices and social media platforms to support interaction, researchers aim to create a dynamic and interconnected learning environment that helps students build and expand their vocabulary knowledge in a meaningful and relevant way [2,34]. From the connectivism perspective, scaffolding plays a role in supporting foreign language learning. It has been interpreted as social assistance benefiting learners: they scaffold one another as they take part in interactive activities and such interactions lead to co-construction of linguistic knowledge [35]. Seamless learning platform provides an essential context that scaffolds students' language learning as it is delivered in multimodal forms, including discussion forums, online feedback mechanisms, and multimedia content sharing tools, which enhance the learners' perceptions and supported social learning and engagement [1,36,37]. [1]adopted the MyCLOUD (My Chinese Language ubiquityOUs learning Days); a social media platform that involves a long-term school-based intervention to scaffold learners vocabulary learning and communicative writing activities under a seamless learning framework.

A schema for English language learning in seamless learning contexts was proposed by [3], highlighting interaction as the central focus and grounding it in connectivism theory, situated learning, and constructivism. Contextualization, social interaction, and continuity were identified as the three essential components of successful language learning by [22]. Building

on previous studies [3,21], this study proposes a theoretical framework that underpins SCVL research in the context of Chinese vocabulary learning, as illustrated in Fig 1.

To note, most reported empirical studies on seamless L2 learning have utilised pretest-posttest designs [12] or focused on describing students' learning processes and investigating the usefulness of developed mobile apps or learning systems [1,13]. These studies have some limitations in their research design, as they may have missed important insights into students' learning outcomes over time and the factors that impact these outcomes, including differences in Chinese language proficiency and inherent motivation and attitude to learning of a diverse learner body. Hence, the use of seamless learning to facilitate CFL vocabulary teaching and learning has remained largely unexplored.

Given these constraints, this study represents a groundbreaking effort in the field by employing a longitudinal DBR intervention that centres on students' autonomous creation of artefacts over an extended duration. Additionally, it integrates an intra-group comparison to explore the trajectory of students' productive vocabulary development in writing in a Chinese higher educational institution context from both qualitative and quantitative approaches. It is worth noting that students are encouraged to employ target vocabulary (i.e., HSK Level 4 words) across diverse contexts and may incorporate specific vocabulary in particular situational contexts. The affirmative and purposeful utilisation of specialised vocabulary, when contextually appropriate, is regarded as a valuable outcome, aligning with [4]'s vocabulary

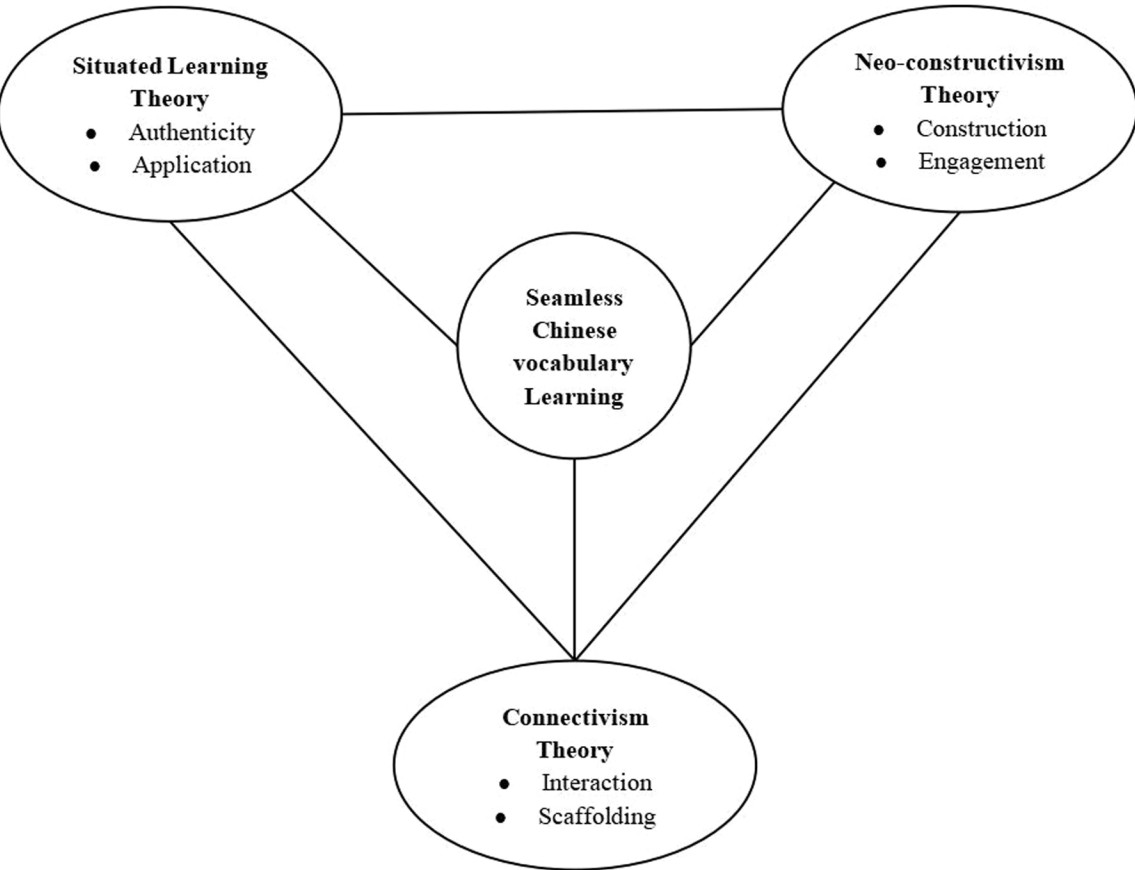

**Fig 1. Theoretical framework in sustaining the research of seamless Chinese vocabulary learning.**

acquisition theory and the notion of seamless learning. By leveraging the artifact-making writing process, the principal aim of this study is to enrich learners' vocabulary growth in Chinese writing, as evidenced by their generated Chinese artifacts, while simultaneously offering valuable insights into the domain of seamless CFL learning.

In summary, the two research questions for the current study are:

RQ 1: To what extent does the utilisation of Seamless Chinese Vocabulary Learning (SCVL) impact CFL HSK Level 4 vocabulary learning effectiveness?

RQ 2: How do learners perceive the use of Seamless Chinese Vocabulary Learning (SCVL) in their CFL HSK Level 4 vocabulary learning?

## Methodology

This study utilised the DBR methodology [19] as its primary research approach for gathering both quantitative and qualitative data. As described by [19], DBR is a flexible empirical data collection method rooted in the learning sciences educational movement pioneered by educational researchers such as Brown and Collins in the early 1980s. DBR endeavours to enhance the scientific comprehension of learning by crafting and executing learning innovations and activities within authentic contexts, aiming to enhance instructional methodologies. Therefore, a mixed method research design was adopted in this study to ensure the robustness and validity of the data collected from participants, each possessing unique characteristics [13,18].

### Research context and participants

The study comprised 32 international students, consisting of 10 males and 22 females, all concurrently enrolled in tertiary education, with an average age of 21.5 years. Participants for this study were purposively selected from intermediate-level Chinese courses at a university (denoted as University A), located on the Chinese mainland, in 2022. They were drawn from different Chinese language classes within the same university to ensure a diverse but comparable representation of learners in the seamless learning program.

The participants for this study were recruited from March 1, 2022 through March 18, 2022. Following the approval of the study's ethical consideration in March, 22, 2022, the study began on March 25, 2022, which was the first day of the semester until it ended on Friday, July 22, 2022. At the time of the study, they were preparing for the HSK Level 4 examination, having already attained proficiency at HSK Level 3. Thus, they demonstrated competency in Chinese Mandarin, capable of communication, discussion, and participation in vocabulary learning activities, particularly within the realm of social media platforms.

A total of 32 participants were involved and all of them were university students who signed the written informed consent form. The study involved the students in an SCVL experience throughout the period of four months between March 28, 2022 to July 15, 2022. All were in a comparatively homogeneous HSK Level 4 course, following the same curriculum and syllabi, and similar to other international students. This reduces heterogeneity in the subjects' variables and makes them consistent in the educational context of the sample. Additionally, participants claimed they had significant usage frequency across various social media platforms (Facebook, WeChat, etc.) and mobile devices confirming the saturation of network technologies in their daily lives as noted by [2].

To further categorise the participants based on their academic performance, their scores on the HSK Level 3 exam (corresponding to CEFR level B1) and Chinese writing grades were obtained from the university placement exams. Therefore, a mixed-ability group of 32 international students with high (i.e., students who scored 290 out of 300 on the HSK Level

3 exam and 90% on the university placement writing assessment) to low (i.e., students who scored 180 out of 300 on the HSK Level 3 exam and 65% on the writing assessment) academic performances in Chinese language participated in the study. All 32 participants were enrolled in a homogeneous HSK Level 4 course, following the same curriculum and syllabus, while also sharing similar international student backgrounds and age range, ensuring a consistent educational background among the participants.

The participating teacher, with four years of teaching experience, was the Chinese teacher for the class. Prior to the intervention, a briefing session was conducted to obtain the support of the policymakers at the university and the consent form were collected from the 32 participants.

## Learning materials, process, and environment design

All participating students in this study were instructed to learn a standardised list of 600 Chinese words, specifically the target HSK Level 4 words, which are commonly utilised in daily communication. The 240 words (i.e., target HSK Level 4 words) were selected from the official HSK Level 4 vocabulary list of 600 words based on four criteria: relevance to students' daily lives and academic contexts, diversity across vocabulary categories, complexity and usage, and alignment with the themes of the learning tasks. For example, words like "广告" (advertisement) and "堵车" (traffic jam) were chosen to encourage authentic usage in sentence-making and essay-writing activities.

Additionally, the selected words were curated from the HSK Level 4 syllabus, following the guidance of three expert teachers with over 10 years of experience in teaching CFL. The word selection process also adhered to the "Syllabus of Chinese Language Program for Foreign Learners in Colleges and Universities" (高等学校外国留学生汉语教学大纲, 2002).

This meticulous word selection process aimed to establish a consistent and standardised vocabulary foundation for all CFL learners, facilitating uniformity and comparability in the research study. The full word list can be found in the S1 Appendix for reference.

The core concept of seamless learning guided the development of the learning model for seamless Chinese vocabulary acquisition in this study. The SCVL framework facilitated learners' language acquisition through two primary dimensions: the seamless Chinese vocabulary learning dimension and the "sentence-paragraph-essay" artifacts making dimension (Fig 2). This model provided a comprehensive framework that integrated the aspects of vocabulary learning and language usage, enabling learners to achieve a holistic understanding of the language and its practical applications.

As shown in Fig 2, for the first dimension of SCVL, mobile phones were used as the learning tools to bridge and interweave formal learning within the classroom (learning engagement and consolidation) with informal learning beyond the classroom (individualised contextual learning, and online peer feedback and interactions). The intention was to nurture the habit-of-mind of self-directed learning in students. For the second dimension of 'sentence-paragraph-essay' artifacts making, the learning content of DingTalk was primarily used in demonstrating the proliferation process of vocabulary knowledge. This was done by promoting social interactions in the DingTalk platform and culminated in the holistic development of language proficiency and achieve vocabulary growth in writing. The SCVL process was a repeated and non-linear learning process, which consisted of the following four activity types:

1. **In-class learning engagement (Activity 1):** These activities were facilitated by the teacher during formal lessons, such as identifying and learning new vocabulary from HSK Level 4 textbook passages. Students engaged in discussions to explore word meanings and usage, observed and mimicked the teacher and their peers, and participated in paired or group

exercises to practice vocabulary. These activities aimed to prepare students for subsequent activities (2) and (3).

2. **Personalised contextual learning (Activity 2):** Individual students, encouraged by the instructor, created and shared photo(s) and texts on social media related to their daily encounters, frequently using the target vocabulary.

3. **Online peer learning (Activity 3):** Students conducted peer review and/or online interactions in the online social network by responding to social media created during activities (1) and (2).

4. **Learning consolidation (Activity 4):** Students returned to formal learning spaces for learning consolidation. The learning consolidation also appeared during online peer learning.

In summary, the learning of vocabulary began with the teacher's input. This was followed by contextualised learning of words through real-life situations, which led to the authentic use of vocabulary learned. This gradually extended, in a bottom-up manner, to the writing of sentences, paragraphs and ultimately essays.

Next, to implement seamless learning of the Chinese language as a foreign language, the DingTalk platform was used as the main tool for facilitating seamless learning. DingTalk, a versatile social media platform compatible with Android, iPhone, Mac, and Windows operating systems, served as the techno-pedagogical model that bridged formal language learning in the classroom with real-life language application and reflection [13]. In this research, each learner was equipped with an individual mobile device on a one-to-one basis, granting them uninterrupted access to the platform both inside and outside the classroom, thereby supporting their language learning endeavours.

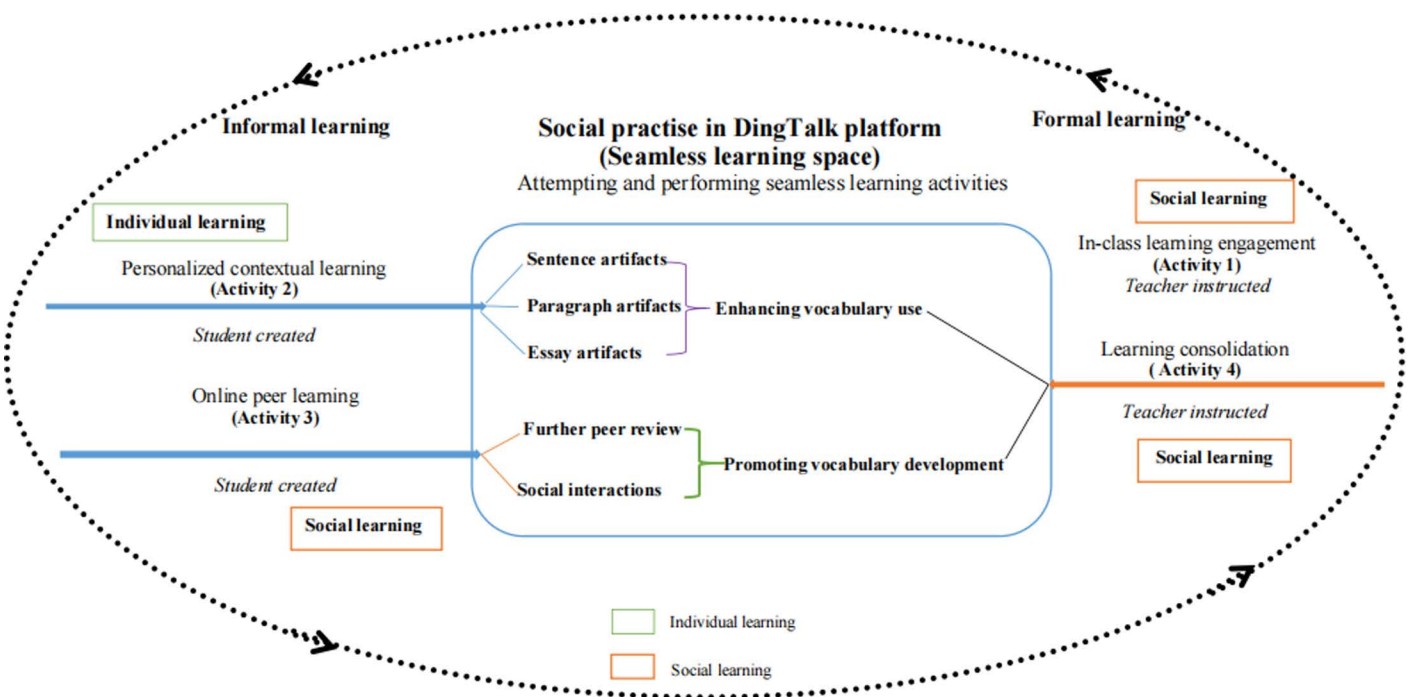

**Fig 2. SCVL framework (Adapted by [1]).**

Moreover, individual DingTalk accounts were provided to engage the learners in SCVL activities. The learners utilised their mobile devices to create artifacts comprising of sentences made using the target vocabulary, subsequently sharing their daily learning experiences on the HSK Level 4 Learning Circle of the DingTalk platform.

The Learning Circle of DingTalk provided the learners with a learning space that enabled them to share their daily learning experience in the interest of informal interactions among the students, teachers, and peers. Explicitly, students could select their designated area of interest. Subsequently, they could post their personal photos and phrases online and communicate with others in Chinese (Fig 3).

Within the Learning Circle, the Online Peer Learning activity (Activity 3) provided each member with the opportunity to comment on and "like" others' posts, thereby fostering a collaborative and supportive learning environment. The availability of peer feedback was ensured, and regular discussions were held to promote continuous learning. Peer feedback was consistently accessible, complemented by regular discussions aimed at fostering ongoing learning. This dynamic learning circle significantly contributed to vocabulary acquisition and effectively supported learners' vocabulary knowledge enhancement in Chinese writing proficiency.

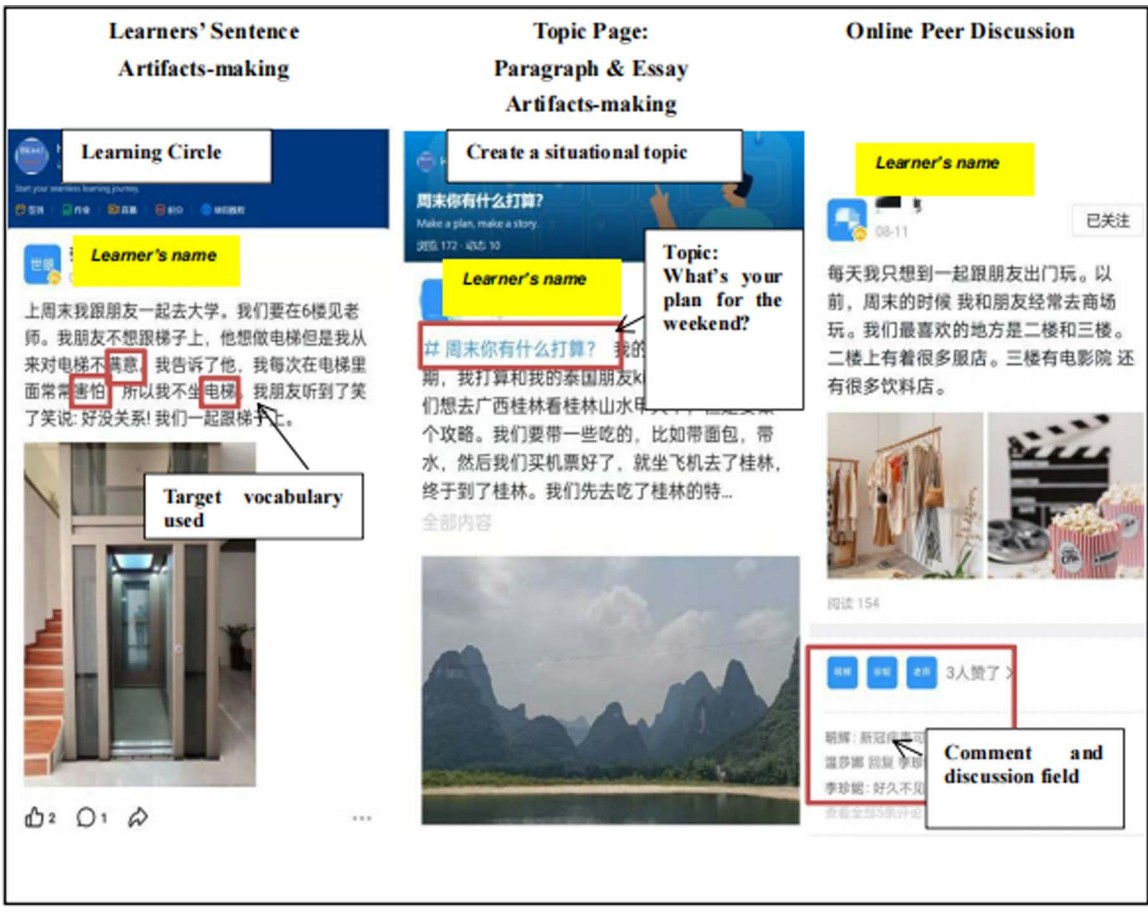

**Fig 3. Screen captures of the main components on the learning circle of DingTalk.**

## Procedure

This study was conducted in March 2022 and was completed within a four-month semester period. The aim was to promote student vocabulary learning in writing through the delivery of a seamless Chinese HSK Level 4 course. The intervention was retrospectively divided into four stages, which could be seen as four mini-cycles of the DBR, with each stage lasting four weeks. During each stage, students focused on making sentence artifacts every week, and after every two weeks, they were allowed to make a paragraph artifact. After completing four weeks or a stage, students were required to make an essay artifact.

There were two learning sessions each week, covering two hours of formal lectures and one hour of informal lectures. During each learning session, the instructor allocated 40 minutes of classroom time to the teaching of new vocabulary (Activity 1), according to the weekly lesson plan for HSK Level 4 syllabus based on the *[Standard]* textbook, to prepare students for subsequent outside-the-classroom contextualized learning activities. With the target vocabulary learned in every session in the formal classroom, students took photos or found suitable pictures, created sentence artifacts, and posted them onto DingTalk, the social media platform (Activity 2). In the next step, students actively commented, corrected, and reviewed each other's artifacts posted in an online peer review to achieve online peer learning (Activity 3). At the end of the learning cycle, the students returned to formal learning of 20 minutes in each session for learning consolidation (Activity 4). With the ongoing SCVL process and weekly learning consolidation, learners further enhanced their vocabulary knowledge.

## Data collection

**Learning circle of DingTalk.** Throughout the intervention, multiple types of data as presented in Fig 3 were collected by the Learning Circle of the DingTalk platform. However, for the research purpose, the analysis in this paper focuses mainly on the student essay artifacts, which was created using target HSK Level 4 vocabulary. 128 essay artifacts were collected for further analysis.

This study was based on a longitudinal intervention with the emphasis on the students' self-directed artefact creations over time and horizontal comparison between groups; therefore, it was not possible to find a control class to make comparison. Also, this study lacked control group comparisons which is in line with DBR philosophy [38]. Consequently, we adopted single group design with intra-group repeated measurements of key outcomes across the time among different groups.

**The SCVL questionnaire.** A questionnaire was designed and administered to assess participants' perceptions of using SCVL for learning Chinese vocabulary on July 18, 2022. The development of the questionnaire was informed by relevant studies and tailored to align with the research questions and theoretical framework of the current study. Two dimensions were investigated: (1) items related to learning experience [16,39]; and (2) items pertaining to self-efficacy and motivation [39,40]. Following iterative consultations with two experts in educational technology and CFL education, minor revisions were made to the initial version. The finalised SCVL learning questionnaire comprised eight questions. All items were rated on a Likert scale ranging from 1 (Strongly Disagree) to 5 (Strongly Agree). The questionnaire exhibited strong internal consistency, with a *Cronbach's α* value of 0.898, indicating high reliability.

**Semi-structured interviews.** Semi-structured individual interviews were conducted with six CFL learners from three different performance groups. These learners were purposively selected based on their HSK Level 3 scores and Chinese writing grades to represent

different proficiency levels: two (2) reflecting low performance, two (2) reflecting medium performance, and two (2) reflecting high performance. The aim was to validate the individual impact of SCVL and to gather in-depth qualitative data on learners' reflections, comments, and perceptions of SCVL, thereby exploring their experiences in greater detail [12]. All of whom actively participated in at least one semi-structured interview (online, 15-20 mins) across the 16 weeks.

The interviews were conducted every two weeks and were guided by a set of questions developed based on the research questions, the participants' SCVL process, and their responses to the questionnaire. The purpose of these interviews was to track and deeply explore students' real-time experiences, emotions, and interactions with the SCVL program. During these interviews, Mandarin was used as the medium of communication, and subsequently, we transcribed and translated the recorded interviews. The demographic information of the six interviewees is listed in Table 2.

## Data analysis

The current study employed a one-sample t-test to determine the overall perceptions of participants towards SCVL, with an alpha level of significance set at 0.05. The mean score of each item was calculated on a scale from 1 to 5. A cut-off point of 3.5 (equivalent to 50%) was used, as suggested by [41,42] If the mean score was above 3.5, it indicated a positive experience and a high perception of the utilisation of SCVL. Additionally, a two-way repeated measures ANOVA was used to analyse the values of vocabulary use in learners' essay artifacts among the three performance groups over time.

Next, after the study, we copied the essay artifacts that had been created and posted by the participants on the DingTalk platform over the four stages of the study into a Microsoft Excel file for further analysis. The Lexical Frequency Profile (LFP) was used to measure vocabulary distribution and showcase vocabulary growth in the texts. The LFP, developed by [43], offers a powerful method to measure productive vocabulary growth. It counts the number of word tokens in a text and distributes them among four frequency bands derived from standardised word frequency lists: the first 1,000 and the second 1,000 most frequent word families, academic words, and other (low frequency) words. For example, if a writing text has 200 words, with 150 in the first 1,000 words, 20 in the second 1,000 words, 20 in the learned vocabulary, and 10 not in the above categories, the LFP for this work would be "75%-10%-10%-5%". This forms a frequency profile that reflects the level of vocabulary use by learners.

In this study, an adapted Chinese LFP was developed by referencing [43] and utilising the HSK 2.0 Word List as its content. This adaptation mirrors the structure of the English LFP but replaces its word family-based approach with a categorisation system aligned with the HSK 2.0 Word List. As shown in Table 1, the HSK 2.0 Word List divides vocabulary into six levels of increasing difficulty, ranging from HSK Level 1 (150 words) to HSK Level 6 (5,000 words).

**Table 2. Demographic information of the interviewees (n = 6).**

| Pseudonym | Age | Gender | Nation | HSK Level 3 Scores | Chinese Writing Grade | Performance Group |
|---|---|---|---|---|---|---|
| MY | 20 | Female | Egypt | 196 | B | Low |
| LT | 20 | Female | Colombia | 198 | C | Low |
| BY | 22 | Female | Thailand | 246 | B | Medium |
| MWD | 21 | Male | Sudan | 251 | B+ | Medium |
| YJW | 24 | Male | Vietnam | 289 | A | High |
| HF | 24 | Male | Colombia | 283 | A | High |

This adaptation addresses the limitations of traditional English LFPs in analysing Chinese texts by accommodating the unique characteristics of the Chinese language and ensuring relevance to the learners' proficiency benchmarks. The use of this tailored LFP framework allows for a precise and pedagogically aligned analysis of Chinese vocabulary acquisition within the seamless learning context.

Next, the software '汉语阅读分级指难针' (available at https://www.languagedata.net/tester/) was employed to measure the Chinese LFP. This software classifies and analyses the frequency of words at each proficiency level in a given text. Specifically, it generates a detailed breakdown of how many words from different levels are present in the text, providing insights into the diversity and complexity of the vocabulary used. The software has been widely adopted and validated in prior research for its reliability and accuracy [44,45].

Based on the HSK 2.0 Word List (refer to Table 1), HSK vocabulary distribution and its role in vocabulary growth can be uncovered by examining frequency lists. The adapted Chinese LFP refers to the distribution rates of HSK vocabulary. Therefore, in accordance with the nature of this research, the percentage of HSK Level 4 vocabulary distribution in essay artifacts was employed to measure the productive vocabulary growth in Chinese writing. The calculation of the adapted Chinese LFP is shown below:

LFP = Number of HSK Level 4 word tokens/Total number of word tokens ×100% (Adapted based on [44])

Note. LFP refers to the HSK Level 4 vocabulary distribution rate

For example, the essay artifact presented in the text box was written by a CFL student and segmented into words. The segmented text is as follows:

情/是/结婚/的/原因/, 也/是/配偶/之间/的/安全/和/尊重/, 一/个/人/难过/的/时候/, 爱人/会/想/办法/为了/让/爱人/很/幸福/, 也/接受/对方/的/缺点/很/重要/, 因为/每/个/人/都/有/缺点/. 即使/有/问题/, 只要/他们/是/一起/就/会/解决/, 关系/就/会/加强/更多/, 不仅/是/因为/爱/, 而且/是/因为/爱/而/产生/的/理解/.

English translation: Love is the reason for marriage, and it is also the security and respect between spouses. When one person is feeling down, the partner will find ways to make the other person happy. It is also important to accept each other's flaws because everyone has them. Even when there are problems, as long as they are together, they will solve them, and the relationship will become stronger. This is not only because of love, but also because of the understanding that arises from love. (Text from one CFL student's writing artifact).

The segmented text was then analysed using the software '汉语阅读分级指难针'(Languagedata) to calculate the frequency distribution of words at each HSK level (1 to 6). The HSK word distribution report was captured from the software and is shown in Fig 4.

表3：HSK词汇档案

| 词表 | 词数 | 词种数 | 分布（%） | 累积分布（%） |
|---|---|---|---|---|
| 一级 | 14 | 8 | 23.33 | 23.33 |
| 二级 | 6 | 4 | 10.00 | 33.33 |
| 三级 | 8 | 7 | 13.33 | 46.67 |
| 四级 | 13 | 12 | 21.67 | 68.33 |
| 五级 | 2 | 2 | 3.33 | 71.67 |
| 六级 | 1 | 1 | 1.67 | 73.33 |
| 超纲词 | 16 | 14 | 26.67 | 100 |
| 总计 | 60 | 48 | 100 | 100 |

**Fig 4. Screenshot of the "HSK Words Distribution Report" (Source: Languagedata).**

To further illustrate the adapted Chinese LFP (i.e., the percentage of HSK Level 4 vocabulary distribution), Table 3 presents the results, including the total number of word tokens, the number of tokens for each HSK level, and the calculated lexical frequency percentages.

As shown in Table 3, the total number of word tokens is 60, with 13 of these being HSK Level 4 word tokens, resulting in an LFP of 21.67%. Subsequently, a two-way repeated measures ANOVA was conducted to investigate potential differences across the four stages. The data collected were analysed using the Statistical Package for the Social Sciences (SPSS) version 26.0. Mean (M) and Standard Deviation (SD) were calculated to present the results for each research question.

To further corroborate the quantitative findings, a thematic analysis was performed on the qualitative data derived from the interviews. Guided by the research questions and participants' performance and responses in the preceding stages, two experienced CFL teachers and researchers were engaged to apply open coding to the raw data. This was followed by axial coding and selective coding to consolidate the data into two primary themes: (1) Affective Engagement and (2) Issues and Challenges. The inter-coder agreement rates, calculated using Cohen's Kappa index, were 0.86 and 0.87 for the two themes, respectively. These values indicate a high level of consistency in the coding scheme [46].

## Results

### Effectiveness of seamless Chinese HSK Level 4 vocabulary learning

To answer RQ 1, the distribution of HSK Level 4 vocabulary in essay artifacts was measured using the adapted Chinese LFP and expressed as a percentage. To examine the statistical significance of the adapted Chinese LFP, a two-way ANOVA was conducted on 128 written texts across all stages. The results of the two-way ANOVA examination are presented in Table 4. However, only the comparisons of the percentages of HSK Level 4 words are presented as one of the main purposes of the research is to encourage the learners to be more inclined to present their thoughts and experiences using the target vocabulary learned (i.e., HSK Level 4 words).

As shown in Table 4, the findings indicate that both the time main effect and group main effect were significant. The time main effect revealed a significant improvement in learners' HSK Level 4 vocabulary distribution across the four stages. Specifically, the LFP increased significantly from 8.67% to 15.57% ($F = 141.818$, $p < 0.001$).

The group main effect was also significant when comparing the three different performance groups. The LFP for the high performance group was significantly higher than

**Table 3. Word distribution report.**

| HSK 2.0 Word list | Number of Word Tokens | Number of Word Types |
|---|---|---|
| HSK Level 1 | 14 | 8 |
| HSK Level 2 | 6 | 4 |
| HSK Level 3 | 8 | 7 |
| HSK Level 4 | 13 | 12 |
| HSK Level 5 | 2 | 2 |
| HSK Level 6 | 1 | 1 |
| Super class words | 16 | 14 |
| In total | 60 | 48 |

HSK Level 4 vocabulary distribution rate: LFP = Number of HSK Level 4 word tokens/Total number of word tokens × 100%

LFP= 13/60 × 100% = 21.67%

**Table 4. LFP for three performance groups across the four stages.**

| Indicator (*M ± SD*) | Group | Stage 1 | Stage 2 | Stage 3 | Stage 4 | Group Main Effect (*F/p*) | Time Main Effect (*F/p*) | Interaction Effect (*F/p*) |
|---|---|---|---|---|---|---|---|---|
| LFP | Low | 5.87% ± 2.17% | 8.28% ± 2.46% | 10.10% ± 2.92% | 12.51% ± 2.48% | 7.634**/ **0.002** | 141.818***/ **<0.001** | 1.144/ 0.345 |
| | Medium | 9.63% ± 2.63% | 11.41% ± 2.80% | 13.13% ± 3.01% | 15.88% ± 2.91% | | | |
| | High | 10.24% ± 2.42% | 11.99% ± 3.73% | 14.38% ± 3.42% | 18.05% ± 3.90% | | | |
| | Total | 8.67% ± 3.07% | 10.63% ± 3.38% | 12.61% ± 3.52% | 15.57% ± 3.83% | | | |

Note: ** p < 0.01, *** p < 0.001.

that for the medium performance group, and the LFP for the medium performance group was significantly higher than that for the low performance group ($F$ = 7.634, $p$ < 0.01). Overall, the learners' LFP presents a consistent trend over time in different performance groups (Fig 5).

However, the interaction effect was not significant ($F$ = 1.144, $p$ > 0.05), which indicates that the outcome of the intervention on LFP was consistent across all performance groups. Thus, whether learners are in the low or high performance groups, the four stages of SCVL benefitted them.

Next, to investigate the impact of SCVL on productive vocabulary growth in Chinese writing using HSK Level 4 vocabulary distribution, the aforementioned findings were supported by specific examples. Specifically, the performance of six learners in terms of LFP from stages 1 to 4 was tracked and analysed to gain a deeper understanding of their CFL writing progress.

Fig 6 presents the learning performance of six tracked learners in terms of LFP results across stages 1 to 4. The results depicted in Fig 6 provide strong evidence of the positive impact of SCVL on individual learners' productive vocabulary growth in terms of LFP over time.

## Participants' perceptions and experiences with the usefulness of SCVL

To answer RQ 2, the quantitative findings on the research questionnaire based on the one-sample *t*-test is shown in Table 5.

As depicted in Table 5, the mean score for the SCVL questionnaire was 4.12, with a standard deviation of 0.43, which was significantly higher than 3.5. This indicates that the participants' affectivity, self-efficacy and motivation with SCVL were notably positive and favourable ($P$ < 0.001).

To further investigate the participating students' affective experiences, self-efficacy, and motivation within the SCVL environment, the findings reveal that students found it meaningful to connect their in-class learning with out-of-class learning through the use of SCVL (Mean = 4.22), as shown in Table 6.

This connection between learning contexts was supported by qualitative data. For instance, one learner commented:

*"The seamless integration of classroom learning with real-life applications through SCVL made learning Chinese more engaging and meaningful."* (L5T, Low, Week 4)

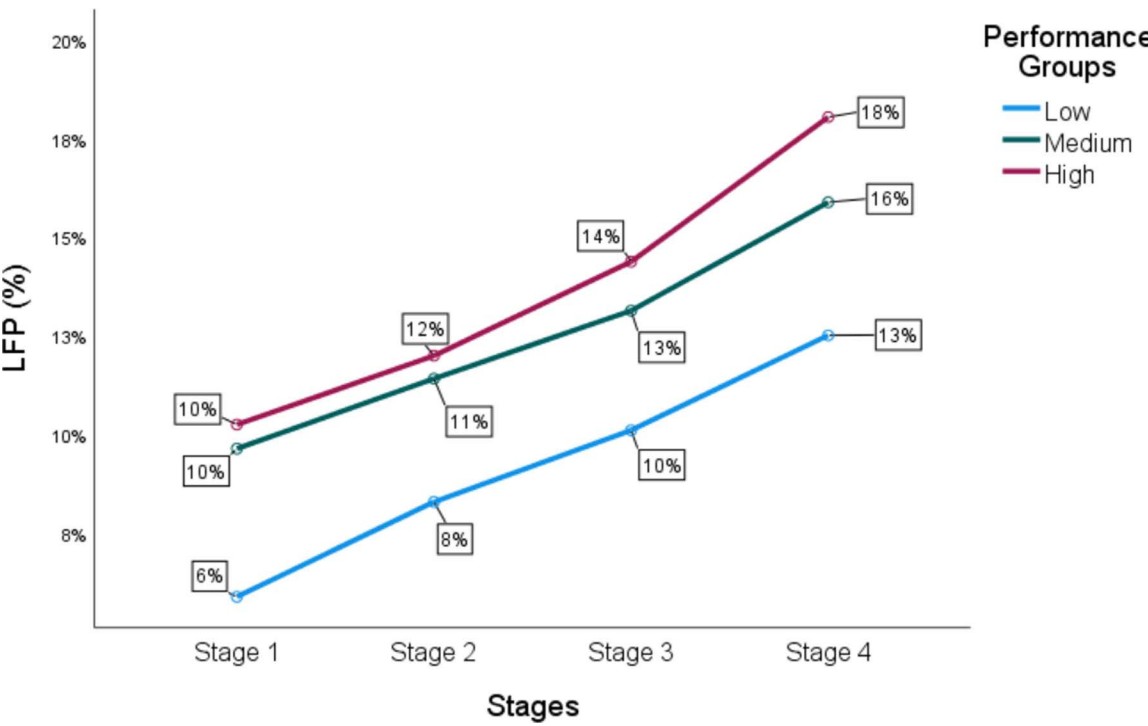

**Fig 5. LFP for all three performance groups across four stages.**

Moreover, students expressed comfort with learning Chinese HSK Level 4 vocabulary through mobile devices outside the classroom, whether at home, on campus, or in other locations (Mean = 4.13). A learner noted:

> *"I feel very comfortable posting Chinese in the HSK Level 4 Learning Circle and interacting with my classmates. I really enjoy it."* (L6T, Medium, I, Week 8)

Similarly, a medium-performing learner shared a comparable experience:

> *"At this stage, I feel much better. We can study by ourselves or discuss in the Learning Circle, which is very relaxed and casual."* (L15T, Medium, F, Week 8)

Furthermore, many participants reported that guided seamless learning enhanced their confidence and motivation in learning Chinese vocabulary (Mean = 4.16), as illustrated in Table 7.

Most students confirmed that the guided informal learning under the seamless Chinese HSK Level 4 vocabulary learning after class hours was important for them to improve the mastery of Chinese vocabulary. HF described that:

> *"Seamless learning is very supportive of my HSK Level 4 vocabulary learning! Every time I go outside with my phone to take pictures and find a scene to create artifacts, I feel like I'm on an adventure. I will think about how to create one interesting artifact that is very stimulating."* (L3, Low, OQ)

Also, many participating students reported that guided seamless learning gave them the confidence and motivation to learn Chinese vocabulary (Mean = 4.16). This was supported

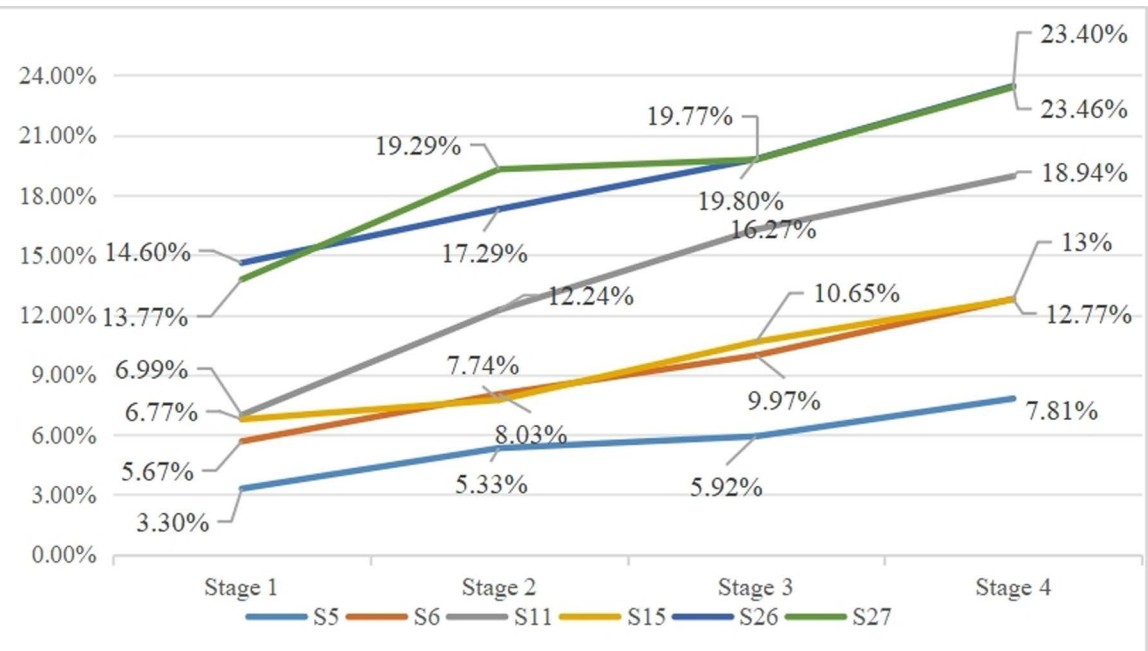

**Fig 6. Six tracked CFL students' LFP from stages 1 to 4.**

**Table 5. Learners' overall perceptions towards the usefulness of SCVL by one sample $t$-test.**

| Mean±SD | $t$ | df | Sig. (2-tailed) |
|---|---|---|---|
| 4.12 ± 0.43 | 14.13 | 29 | **0.000** |

**Table 6. Results of the items on experience in SCVL.**

| Item | Mean score | SD |
|---|---|---|
| I am comfortable with Chinese HSK Level 4 vocabulary learning through my mobile device outside of the classroom (at home, on campus, and other places, etc.). | 4.13 | 0.70 |
| It is interesting to learn Chinese HSK Level 4 vocabulary under seamless learning after class hours. | 3.94 | 0.83 |
| I am happy to share content in Chinese about my daily life on social media platform. | 4.03 | 0.85 |
| It is meaningful to connect my in-class learning to my out-of-class learning of Chinese. | 4.22 | 0.60 |

**Table 7. Results of the items on self-efficacy and motivation in SCVL.**

| Item | Mean score | SD |
|---|---|---|
| I am able to learn Chinese HSK Level 4 vocabulary from in class formal learning to out-of-class informal learning through the use of mobile devices and social media platforms. | 4.03 | 0.64 |
| Informal learning guided by teaching instructor in seamless Chinese HSK Level 4 vocabulary learning can support my formal classroom learning. | 4.25 | 0.71 |
| Guided informal learning under seamless Chinese HSK Level 4 vocabulary learning after class hours is important for me to improve the mastery of Chinese vocabulary. | 4.19 | 0.80 |
| Guided seamless learning gives me confidence and motivation in learning Chinese vocabulary. | 4.16 | 0.76 |

by qualitative data where the participants were more confident to learn Chinese by posting artifacts and improved their self-perception from self-doubt into self-confidence. This is illustrated in the following quotation:

> *"I enjoyed the process of seamless Chinese vocabulary learning, especially when my teachers and classmates upvoted my artifact with a 'like', which made me become self-confident in Chinese writing."* (L4, Low, I, Week 16)

> *"I feel like my self-confidence has improved a lot since I started the program. I used to doubt my abilities, but now I feel more capable and confident in what I can accomplish. I also feel that I have significantly improved my Chinese writing skills, as I am now able to construct longer sentences and learn independently."* (L17, Medium, I, Week 16)

Another student said that he learned a lot of new words, which made him more and more motivated.

> *"First of all my vocabulary has really improved, every time I take a picture or do a seamless learning activity I learn new vocabulary unexpectedly, last time I took the bus to the mall I had to use the word traffic jam, I not only used the word traffic jam but I also learned the word '加塞 congestion and the word '头盔 helmet' for a motorbike I also learned the word 'helmet' for motorbikes. My teacher complimented me on the way I learned. I feel more and more motivated."* (L27T, Medium, OQ)

Moreover, it is apparent from the word cloud that learners felt SCVL was helpful to their Chinese learning (Fig 7). Correspondingly, the findings revealed that most of the learners assuredly gained various benefits from experiencing seamless learning study utilising SCVL.

Among the issues and challenges reported by learners include the noisy formal learning environment, internet connectivity, distractions, and technical issues in the formal learning setting, as highlighted in the following.

> *"I experienced a weak internet connection during the in-class learning sessions." (L5, I, Week 2, Medium)*

> *"My internet keeps disconnecting. I can't do any seamless learning activities or practice vocabulary due to my poor internet connectivity."* (L29, I, Week 2, High)

Three learners expressed difficulties in maintaining focus during formal learning, as indicated by the following quotation:

> *"I'm constantly distracted by other things, such as eating, drinking, texting, or surfing the web for other purposes."* (L22, I, Week 4, Low)

Specifically, some learners highlighted a drawback of using the seamless Chinese learning circle, which is the inability to withdraw a posted artifact. One learner provided feedback regarding this issue, expressing the following sentiment:

> *"Why can't I delete my posted artifacts from the learning circle of DingTalk platform? I have to delete it and redo one."* (L11, F, Week 4, Medium)

Two learners expressed their difficulty in completing the essay writing task within the one-hour time limit in the fourth week. As one learner described it:

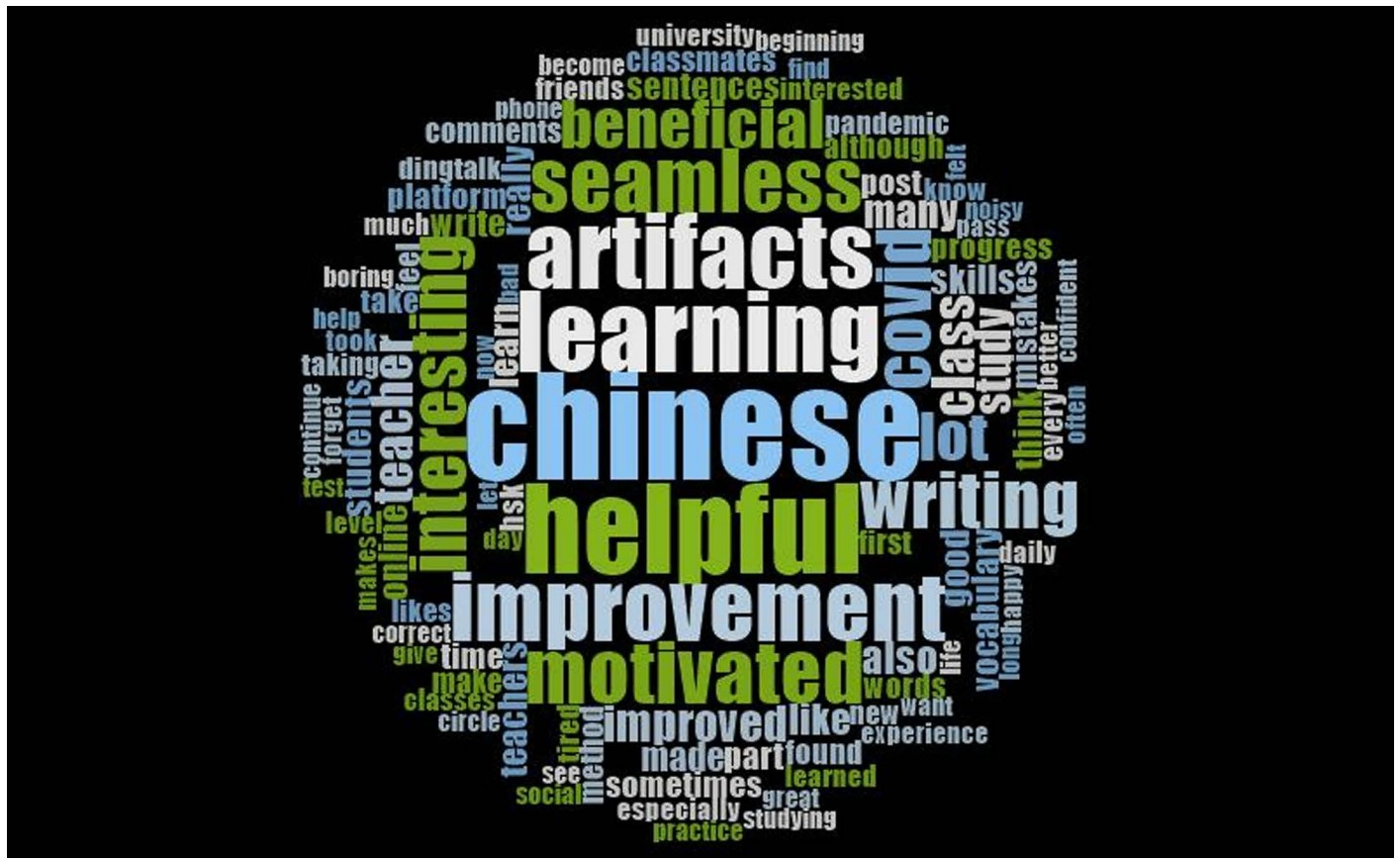

**Fig 7. Word cloud created by NVivo 12 Plus (Data source: Open-ended questionnaire).**

*"I was fearful of Chinese writing and can't manage to put words on the screen within the one-hour time limit."* (L2, I, Week 4, Low)

*"It's hard for me to manage the time to finish the essay and then post it in the 'Homework' function."* (L6, I, Week 4, Low)

In a nutshell, although there were several issues and challenges encountered in the SCVL environment, the learners valued their experiences and undoubtedly gained benefits.

## Discussion

Seamless learning, with its emphasis on integrating formal and informal learning environments, offers a dynamic framework for addressing challenges in second language acquisition [2,13]. Through the four-stage SCVL, this study found that the SCVL framework via the "sentence-paragraph-essay" artefact-making writing process was effective in promoting CFL students' vocabulary enhancement in the LFP for the high, medium, and low performance groups in essay writing. This aligns with prior research. Learners' LFP for vocabulary growth in a seamless CFL learning environment showed a positive increasing trend over time, as emphasized by [2]. The current research reported a substantial difference in the LFP of CFL learners based on the distribution of HSK Level 4 vocabulary in their essay artifacts. These variations, according to [47,48], should be regarded as strong indicators of

learners' productive vocabulary growth in writing. This may be attributable to the nature of continuous learning, contextualisation, flexibility, and integration in the SCVL used in this research. Seamless learning enables learners to connect their in-class learning to out-of-class learning without any boundaries or by leveraging technology to support learning across different contexts and environments [22,49]. In this study, the learning of vocabulary begins with the educator's input. This is followed by the contextualised learning of words outside of the classroom, which leads to the authentic use of vocabulary learned in various contexts. This gradually extends, in a bottom-up manner, to the writing of sentences, paragraphs, and ultimately essays. The findings support [33]'s connectivism theory, where seamless learning is not bound by any specific location but rather consists of networks of connections constructed from experience and interactions among learners, technologies, and their surroundings [1,50]. The application of the connectivism theory to SCVL highlights the significance of networks and technology in fostering ongoing and connected learning [13,51]. In this study, four interactive seamless learning activities were implemented through the use of mobile devices and DingTalk platform to facilitate learner interaction. The aim was to establish a dynamic and interconnected learning environment that could assist learners in expanding their vocabulary knowledge in a relevant and meaningful way.

Overall, the findings suggest that long-term vocabulary knowledge building practice has a positive effect on the development of productive vocabulary growth [48,49]. As elaborated in section of 'learning materials, process, and environment design', a number of SCVL activity designs may have contributed to learners' productive vocabulary knowledge building. These included in-class learning engagement; contextualised sentence-making and photo-taking; online peer learning; and learning consolidation. As an example, through online peer discussion, learners browsed and commented on each other's artifacts, including those posted by learners whose Chinese proficiency was higher than their own, facilitated by a mix of learners with different levels of HSK 3 Chinese proficiency. During peer interaction, they frequently encountered new vocabulary, which they then considered, learned, and applied in their subsequent writing. The finding supported [36]'s assertion that social interactions with other people expose learners to rich and diverse input and output, which is critical. The more frequently a learner encounters a vocabulary item, the more likely he is to learn the word. The integration of different modes of learning (e.g., formal and informal learning, personal and social learning, etc.) involving both language input and output allows for deeper processing of language and knowledge [2,13]. Language use and language learning co-occur through this integrated process, with language use for meaning-making in mediating language learning. Notably, DingTalk is an SCVL platform that aims to foster a mediated social network for the Chinese language learning community [28,36]. It nurtures a social network where language learning and applications are intertwined over time. This lends support to [2,13], who highlighted that operating social networks with lucidity can promote peer feedback and can be a potential device to contribute to knowledge construction. Notably, DingTalk can be regarded as a seamless learning space to bridge activities in and outside of the classroom that learners can use to organise their learning experiences [36].

Regarding learners' HSK Level 4 vocabulary distribution among the three performance groups, the LFP for the high performance group was significantly higher than the medium performance group, and the LFP for the medium performance group was higher than the low performance group.

When comparing previous research, this finding is in line with [43]'s study on the same topic in investigating LFP in English-language essays. As many scholars believe, the higher the language competence, the more vocabulary will be produced [48,52]. This finding is interesting given the fact that in using HSK Level 4 words in their writing, as discussed

previously, all CFL learners with mixed Chinese proficiency levels attended the seamless Chinese HSK Level 4 learning course and were encouraged to apply them to their daily lives. However, when it comes to the production of HSK Level 4 words, the results were quite different. A possible explanation is that many learners in the medium group prepared for the upcoming HSK Level 4 test at the end of the semester. This explanation was supported by the qualitative data.

As learners navigate diverse settings and technologies in seamless learning activities, they encounter opportunities for speaking and writing within genuine and authentic scenarios [23]. This aligns with [53]'s notion of language production for feedback and correction. Seamless learning provides an organic backdrop for learners to engage in language application, receive feedback from varied sources, and refine their language output [15,20,21]. Thus, the convergence of seamless learning and Swain's comprehensible output hypothesis supports a dynamic and holistic approach to language acquisition that emphasises active participation, feedback, and language production across diverse contexts.

Next, the one sample t-test revealed that after the SCVL research study, learners gained positive experiences with the use of SCVL. This outcome is similar to the findings of the research carried out by [54] which focused on EFL learners' CALL-based vocabulary learning experience. Positive experiences and affections were recognised as a motivating factor for foreign language learners to be active in the learning process [10,16]. This finding implies that the use of SCVL in learning Chinese vocabulary is likely to make the learners become active learners.

Based on the qualitative data, the majority of CFL learners reported having positive experiences based on two themes: affective engagement, and issues and challenges they encountered while participating in SCVL. The majority of learners expressed positive emotions towards participating in seamless learning activities (e.g., in-class learning engagement and online peer learning). The word cloud in Fig 7 offers a visual representation of the most frequently cited terms in students' reflections and feedback on their experience with SCVL. Prominent words such as "helpful," "improvement," and "motivated" stand out, reflecting recurring themes in the feedback and underscoring students' perceptions of SCVL's positive impact. These results indicate that learners widely recognise the benefits of SCVL in enhancing both their vocabulary acquisition and overall Chinese language learning. Many CFL students highlight how the seamless integration of practice with real-world learning contexts made their study experience more engaging and practical, suggesting that this approach not only supported their learning but also fostered greater motivation [25]. This qualitative insight aligns with the overall positive feedback regarding SCVL's effectiveness in promoting vocabulary development.

However, this research cannot deny the existence of several challenges in implementing SCVL. For example, some learners in the study identified noisy formal learning environments and poor internet connections as common issues. Such concerns were consistent with previous research that identified IT and internet literacy as potential barriers to successful seamless learning implementation [3,13]. Thus, it is imperative for educators to assist learners who may face technological disadvantages in their home environment by ensuring that such learners receive priority access to university internet facilities [30,34].

Hence, this study suggests that the SCVL approach effectively leverages mobile technologies to integrate vocabulary learning with writing activities, such as student essays, which require learners to apply target vocabulary from the HSK Level 4 list in meaningful and contextualised ways. By combining the formal structure of classroom learning with the flexibility of informal, mobile-assisted activities, SCVL demonstrates how seamless learning can be tailored to support specific language acquisition goals [34]. Furthermore, learners reported an

overall enjoyable learning experience, attributing their improved Chinese vocabulary acquisition and writing skills to the benefits of SCVL, despite encountering some challenges along the way.

## Conclusions and implications

This study sought to examine the efficacy of employing the SCVL framework in enhancing vocabulary acquisition among international students in mainland China. Following the DBR framework comprising four iterative stages, the 16-week intervention aimed to foster an engaging and interactive learning environment conducive to fostering productive vocabulary expansion in Chinese writing. The findings revealed that the SCVL framework significantly bolstered CFL learners' vocabulary acquisition, particularly in terms of LFP across various phases of learning and among diverse performance cohorts. These results underscore the importance of in-depth, interactive, and sustained vocabulary development in facilitating notable advancements in Chinese language proficiency [2,8]. To note, the use of SCVL offers several advantages to students in improving their Chinese vocabulary learning for writing, particularly in vocabulary growth. The "sentence-paragraph-essay" artifact making process helps students retain new vocabulary words in their long-term memory.

Notwithstanding the valuable insights provided by this study, there are several limitations that should be considered. Specifically, the small sample size employed in the study, while appropriate for a preliminary investigation, cautions against generalizing the findings to a broader population.

Additionally, the study solely focused on a single group design with intra-group comparison of seamless Chinese HSK level 4 learning, without including control items to investigate potential differences between conventional learning and seamless learning. To address these limitations, future research could investigate the voices of CFL learners to triangulate the data and provide more evidence of differences. This could involve conducting interviews with learners to describe their seamless learning experiences and perceptions, thereby providing a more comprehensive understanding of the impact of the SCVL framework on Chinese language proficiency.

Despite these limitations, this study contributes significantly to the theoretical rationales of technology-supported education. It offers valuable insights for researchers, developers, educators, and students, providing a better understanding of the integration and application of the SCVL model in learning and teaching contexts [12]. Moreover, the findings will be particularly beneficial for educators seeking to engage learners in seamless learning experiences. Encouraging self-directed and social learning through language artifacts rooted in learners' life experiences presents a pedagogically viable approach to implementing seamless learning [55].

However, transitioning between formal (e.g., in-class learning) and informal (e.g., out-of-classroom) learning within the SCVL framework poses notable challenges for CFL learners. One potential solution is the use of learning analytics, which involves collecting and analyzing data on learners' behaviors and performance across different learning environments [22]. This data can inform the design of cohesive learning experiences that seamlessly bridge these environments, thereby enhancing overall learning outcomes [22,24].

This study specifically focused on the use of seamless learning for vocabulary acquisition, excluding areas such as grammar, speaking, or listening skills. As grammar remains a contentious topic in foreign language education [56], and given the motivation to enhance communicative competence in CFL learners, future research should explore the potential of SCVL to support grammar acquisition, as well as listening and speaking tasks [12].

## Supporting information

**S1 Appendix. Target HSK level 4 words.**
(DOCX)

## Author contributions

**Conceptualization:** Xiaosheng Zhou, Ying Soon Goh.

**Data curation:** Xiaosheng Zhou.

**Formal analysis:** Xiaosheng Zhou.

**Funding acquisition:** Xiaosheng Zhou.

**Investigation:** Xiaosheng Zhou.

**Methodology:** Xiaosheng Zhou, Ying Soon Goh.

**Project administration:** Ying Soon Goh.

**Resources:** Ying Soon Goh.

**Software:** Ying Soon Goh.

**Supervision:** Ying Soon Goh.

**Validation:** Ying Soon Goh.

**Visualization:** Ying Soon Goh.

**Writing – original draft:** Xiaosheng Zhou.

**Writing – review & editing:** Xiaosheng Zhou, Ying Soon Goh.

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
