## [Decision Letter · Decision Letter 0]

29 Oct 2024

Dear Dr. Zhou,

We look forward to receiving your revised manuscript.

Kind regards,

Lawrence Jun Zhang, Ph.D.

Academic Editor

PLOS ONE

Journal Requirements:

3. We note you have included a table to which you do not refer in the text of your manuscript. Please ensure that you refer to Table 7 in your text; if accepted, production will need this reference to link the reader to the Table.

Reviewers' comments:

Reviewer's Responses to Questions

**Comments to the Author**

1. Is the manuscript technically sound, and do the data support the conclusions?

Reviewer #1: Partly

Reviewer #2: Yes

2. Has the statistical analysis been performed appropriately and rigorously?

Reviewer #1: Yes

Reviewer #2: Yes

3. Have the authors made all data underlying the findings in their manuscript fully available?

Reviewer #1: Yes

Reviewer #2: No

4. Is the manuscript presented in an intelligible fashion and written in standard English?

Reviewer #1: Yes

Reviewer #2: Yes

Reviewer #1: I think that this is a powerful, informative paper which has the potential to be published and welcomed by the readers of this journal. Overall, the rationale for the paper is clear, the topic, Chinese vocabulary acquisition in higher education, is an area in need of further research, and the authors have designed a learning environment influenced by learning theories, and proven pedagogical learning designs in the form of seamless learning. However, there are some areas that I feel need to be addressed before this study is ready for publication.

The introduction clearly presents the background and some of the aims of the study, but it would be better if you clearly described the main learning tasks and methodology and how they are related to the study. For example, more emphasis needs to be placed on the article writing e.g. authentic, contextual tasks which complement Seamless learning. This needs to be emphasised from the beginning as I feel it is an important aspect of this study. Also, you could mention DBR in the introduction. Perhaps you could discuss why DBR complements seamless learning to strengthen the aims of the study.

The following sentence should be moved from section 3.4 to the introduction to help readers understand the structure of the paper: “However, for the research purpose, the analysis in this paper focuses mainly on the student essay artifacts, which was created using target HSK Level 4 vocabulary.”

The methodology is mainly well-explained and follows proven steps in DBR but there are a few areas that need further explanation. For example, in section 3.2 please explain how the 240 words were selected. What criteria was used to select the words from the list of 600?

Were the students in China at one university or studying overseas? You mention they were selected from universities from all over China but then took the same course.

There is no explanation of why activity 1 in fig 2 is a social learning activity. It is teacher led activity in the classroom but what is the social aspect? Please explain more about the social learning in the classroom because not all classroom learning can be classed as social learning. For me, it is the activity that makes the learning social.

The class is described as a mixed ability class but for me, the definition of mixed ability would be students from different levels e.g. HSK1, 2, and 3 etc. Perhaps the students should be described as high and low scoring because I think all the students are at the same recognised ability level, HSK 3.

When were the students interviewed? Was it after the study or during the study.

What software did you use to measure the Chinese LFP? Without mentioning the software or process it would be hard for researchers to replicate this procedure.

I do not understand what table 3 shows. Is table 3 an example of the written text above table 3. Perhaps a sentence to explain it would make it clearer. Also, the text box above table 3 needs a caption.

The results and discussion are interesting and seem logical. I just have a couple of suggestions.

Figure 5 needs explaining in more detail. Why did you select data from 6 learners and what is the purpose of showing this data.

Please explain the word cloud in figure 6 in terms of results.

There are a few typos that need correcting. For example, the sentence “Figure Error! No text of specified style in document.” and “figure 4.16” which does not exist.

The manuscript needs page numbers which would also make it easier to give you feedback.

I hope my feedback and suggestions will help you improve what I consider to be a very interesting, well-designed manuscript, and I would be willing to review the final draft.

Reviewer #2: This study is well-designed, with an appropriate and comprehensive literature review. However, the Introduction requires improvement in terms of flow. The concepts are presented in a disjointed manner without a clear connection between them. For example, the first paragraph discusses seamless learning, but the second paragraph shifts abruptly to vocabulary without any transitional elements. Similarly, the discussion transitions to the SCVL approach without making the relationship between these topics clear. Improving the clarity and logical flow of these sections will enhance the paper’s readability.

The paper’s overall formatting also needs revision. Issues such as the error message “Figure Error! No text of specified style in document...” and the lack of clarity in the tables and figures detract from the presentation. The authors should consider consulting other published papers for reference to improve the layout and presentation of visual elements. Furthermore, the statistical results need some formatting adjustments, such as italicizing symbols and ensuring appropriate spacing before and after symbols.

There appears to be a significant issue with the reported standard deviations (SD) in Tables 6 and 7, as the ranges are unusually large, suggesting potential errors. Please verify and report the correct values.

Lastly, it is important to clarify the application of The Lexical Frequency Profile (LFP) in this study. While LFP in English vocabulary studies relies on the word family count method, it is unclear whether the Chinese LFP used here is based on a word list organized by word families. Please provide additional details regarding this aspect to ensure clarity.

**Do you want your identity to be public for this peer review?** For information about this choice, including consent withdrawal, please see our Privacy Policy

Reviewer #1: No

Reviewer #2: **Yes: ** Atsushi Mizumoto

---

## [Author Response · Author response to Decision Letter 1]

9 Dec 2024

Dear Reviewer(s) and Editor(s),

We would like to extend our sincere gratitude to the Editors and Reviewers for their insightful comments and constructive feedback, which have significantly contributed to the improvement of our manuscript. We greatly appreciate the time and effort invested in reviewing our work and providing detailed suggestions.

In our "Response to Reviewers Letter," we have included the Reviewers’ and Editors’ comments (formatted in black), followed by our Author’s response (formatted in red), which outlines how the comments have been addressed or provides further explanation. Relevant excerpts from the revised manuscript that demonstrate how specific points have been addressed are included (formatted in blue).

The revised manuscript employs the "Track Changes" feature to clearly identify all corrections and adjustments. Each comment has been thoroughly addressed to ensure clarity and alignment with the high standards of the journal.

We trust that our revisions meet the expectations of the Editors and Reviewers. If there are any remaining questions or additional feedback, we would be happy to address them promptly. Your guidance has been invaluable in refining the quality and rigor of our research.

Thank you once again for your support and consideration.

Kind regards,

Corresponding Author: ***

---

## [Editor Report · Decision Letter 1]

11 Dec 2024

Dear Dr. Zhou,

Thank you for submitting your manuscript to PLOS ONE. After careful consideration, we feel that it has merit but does not fully meet PLOS ONE’s publication criteria as it currently stands. Therefore, we invite you to submit a revised version of the manuscript that addresses the points raised during the review process. This time round, we would like to see a clean version of your paper that is ready for acceptance. So, please read your revised version carefully to endure it is typo-free.

We look forward to receiving your revised manuscript.

Kind regards,

Lawrence Jun Zhang, Ph.D.

Academic Editor

PLOS ONE
---

## [Author Response · Author response to Decision Letter 2]

25 Jan 2025

Dear Reviewer (s) and Editor (s),

First and foremost, we would like to express our utmost gratitude to the Editors and Reviewers for their valuable comments and suggestions and for the effort and time spent in attempting to improve the quality of this article throughout the review process. As such, we have attempted to address all queries and corrections as best as possible.

“Reviewers’ and editor’s comments” have been included (written in black), followed by “Author’s response” (written in red), which explains how the changes have been incorporated or provides further rationale. Some extracts from the paper to show how the reviewers’ comments have been addressed (written in blue). The revised manuscript with tracked changes clearly indicates the location of corrections and adjustments, utilizing the “Track Changes” feature.

We trust we have met the expectations of the Editor and Reviewers. If there are any further concerns or suggestions, please do not hesitate to let us know. Your feedback is immensely valuable to us in improving the quality and rigor of our research.

Sincerely,

Corresponding Author: ***

---

## [Editor Report · Decision Letter 2]

30 Jan 2025

Knowledge Building and Vocabulary Growth: Assessing the Impact of Seamless Chinese Vocabulary Learning for International Students

PONE-D-24-28347R2

Dear Dr. Zhou,

We’re pleased to inform you that your manuscript has been judged scientifically suitable for publication and will be formally accepted for publication once it meets all outstanding technical requirements.

Kind regards,

Lawrence Jun Zhang, Ph.D.

Academic Editor

PLOS ONE
---

## [Editor Report · Acceptance letter]

PONE-D-24-28347R2

PLOS ONE

Dear Dr. Zhou,

I'm pleased to inform you that your manuscript has been deemed suitable for publication in PLOS ONE. Congratulations! Your manuscript is now being handed over to our production team.

Kind regards,

on behalf of

Professor Lawrence Jun Zhang

Academic Editor

PLOS ONE